# BDC-Occ: Binarized Deep Convolution Unit For Binarized Occupancy Network

## Abstract

Existing 3D occupancy networks demand significant hardware resources, hindering the deployment of edge devices. Binarized Neural Networks (BNNs) offer a potential solution by substantially reducing computational and memory requirements. However, their performances decrease notably compared to full-precision networks. In addition, it is challenging to enhance the performance of the binarized model by increasing the number of binarized convolutional layers, which limits its practicability for 3D occupancy prediction. This paper presents two original insights into binarized convolution, substantiated with theoretical proofs: (a) $1 \times 1$ binarized convolution introduces minimal binarization errors as the network deepens, and (b) binarized convolution is inferior to full-precision convolution in capturing cross-channel feature importance. Building on the above insights, we propose a novel binarized deep convolution (BDC) unit that significantly enhances performance, even when the number of binarized convolutional layers increases. Specifically, in the BDC unit, additional binarized convolutional kernels are constrained to $1 \times 1$ to minimize the effects of binarization errors. Further, we propose a per-channel refinement branch to reweight the output via first-order approximation. Then, we partition the 3D occupancy networks into four convolutional modules, using the proposed BDC unit to binarize them. The proposed BDC unit minimizes binarization errors and improves perceptual capability while significantly boosting computational efficiency, meeting the stringent requirements for accuracy and speed in occupancy prediction. Extensive quantitative and qualitative experiments validate that the proposed BDC unit supports state-of-the-art precision in occupancy prediction and object detection tasks with substantially reduced parameters and operations. Code is provided in the supplementary material and will be open-sourced upon review.

## 1 Introduction

Recent advancements in 3D occupancy prediction tasks have significantly impacted the fields of robotics (DeSouza & Kak, 2002; Ye et al., 2024; Lin et al., 2024) and autonomous driving (Shi et al., 2023; Yan et al., 2024; Zhang et al., 2024; Wang et al., 2024), emphasizing the importance of accurate perception and prediction of voxel occupancy and semantic label within 3D scenes. However, occupancy prediction requires predicting dense voxels, which leads to substantial computational expenses (Cao & de Charette, 2022; Wang et al., 2023; Liu et al., 2024). Moreover, the formidable performance of occupancy prediction models relies on increasing model size (Li et al., 2023b). These factors collectively hinder the deployment of high-performance occupancy prediction networks on edge devices. For instance, Convolutional Neural Networks (CNN) (He et al., 2016; Krizhevsky et al., 2017; Ronneberger et al., 2015; Lin et al., 2017) possess hardware-friendly and easily deployable characteristics. Moreover, CNN-based occupancy prediction networks (Huang et al., 2021; Huang & Huang, 2022) exhibit outstanding performance, making them the primary choice for deployment on edge devices. However, high-performance CNN-based occupancy networks (Cao & de Charette, 2022; Li et al., 2023b) often involve complex computations and numerous parameters. Therefore, it is necessary to introduce model compression techniques (Deng et al., 2020) to reduce the computational complexity and parameter count of CNN-based occupancy networks.

Research on neural network compression and acceleration encompasses four fundamental methods: quantization (Gholami et al., 2022), pruning (Liang et al., 2021), knowledge distillation (Gou et al.,

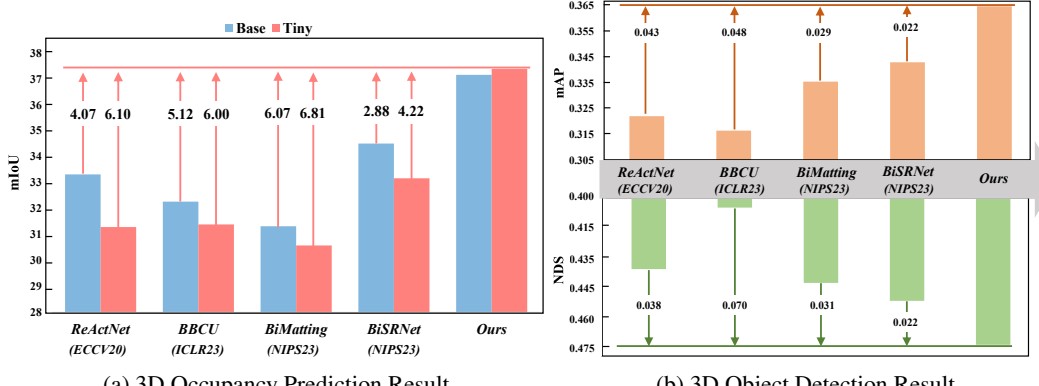

(a) 3D Occupancy Prediction Result        (b) 3D Object Detection Result

Figure 1: **Comparison between our BDC and state-of-the-art BNNs in the 3D occupancy prediction and 3D object detection tasks.** For the 3D occupancy prediction task, Base means binarizing the BEV encoder and occupancy head, Tiny means further binarizing the image neck based on Base. For the 3D object detection task, all binarized models are in Tiny.

2021), and lightweight network design (Zhou et al., 2020). Among these methods, Binarized Neural Networks (BNN), which fall under the quantization category, quantize the weights and activations of CNN to only 1 bit, leading to significant reductions in memory and computational costs. By quantizing both weights and activations to 1 bit, BNN (Hubara et al., 2016) can achieve a memory compression ratio of $32\times$ and a computational reduction of $64\times$ when implemented on Central Processing Units (CPU). Furthermore, compared to full-precision models, BNN (Hubara et al., 2016) only requires logical operations such as XNOR and bit counting, making them more easily deployable on edge devices.

Recent studies, such as BBCU (Xia et al., 2022) and BiSRNet (Cai et al., 2024), have demonstrated the capability of binarizing complex models with promising performance in tasks such as image super-resolution (Yang et al., 2019) and denoising (Tian et al., 2020). We try replacing each full-precision convolutional unit in the occupancy network with the binarized convolutional units proposed by these binarization algorithms. Such a binarized model could achieve a respectable level of accuracy but still a notable performance gap compared to the full-precision model. In full-precision models, it's common sense that increasing convolutional layers can lead to performance improvements. However, the binarized model did not exhibit a trend of performance improvement as the number of binarized convolutional layers increased. Instead, there is a tendency for performance to decline, making it challenging for binarized models to improve performance by increasing the number of convolutional layers (Xia et al., 2022). Insufficient performance of binarized occupancy networks inevitably will have adverse effects on the perception of 3D space, thereby restricting the deployment of binarized models in autonomous vehicles.

Therefore, addressing the issues of decreasing accuracy with increasing binarized convolutional layers and limited perceptual capability is crucial for bridging the performance gap between binarized and full-precision models. To tackle these challenges, we propose a novel BNN-based method, namely Binarized Deep Convolution Occupancy (**BDC-Occ**) network for efficient and practical occupancy prediction, marking the **first** study of binarized 3D occupancy networks. Our novel insights stem from two intrinsic properties of binarized convolution: (a) $1 \times 1$ binarized convolution introduces minimal binarization errors as the network deepens, and (b) binarized convolution is inferior to full-precision convolution in capturing cross-channel feature importance. Drawing on these insights, we limit additional binarized convolutional kernels to $1 \times 1$ to reduce the impact of binarization errors as the network depth increases. Secondly, we introduce a per-channel refinement branch that leverages newly added convolutional layers to narrow the gap with the output of full-precision convolution through first-order approximation. Integrating the two proposed techniques, we develop the **B**inarized **D**eep **C**onvolution (**BDC**) unit, which remarkably enhances binarized model performance, despite the deepening of the binarized convolutional layers. We decompose the 3D occupancy network into four fundamental modules and customize binarization using the BDC unit for each module.

The innovations and contributions of this paper are summarized as follows:

**(i)** Based on the original insights reinforced with theoretical proofs, we propose **B**inarized **D**eep **C**onvolution (**BDC**) unit, further introduce a novel BNN-based occupancy network named BDC-Occ. To our knowledge, this is the first paper to study the binarized occupancy network.

**(ii)** In the BDC unit, additional binarized convolutional kernels are constrained to $1 \times 1$ to minimize the effects of binarization errors as the network depth increases. Subsequently, we propose a per-channel refinement branch to reweight the output via first-order approximation, thereby mitigating the limitations of binarized convolutional layers in assigning importance to features across channels. The 3D occupancy network is further decomposed into four fundamental modules, allowing for a customized design using the BDC unit.

**(iii)** The proposed BDC unit reduces binarization errors and enhances perceptual capability while considerably increasing computational efficiency, thus meeting the demanding requirements for accuracy and speed in occupancy prediction. Extensive experiments on the Occ3D-nuScenes dataset demonstrate that our method achieves state-of-the-art (SOTA) mIOU, closely approaching that of full-precision models while utilizing only **52.26%** of the operations and **59.97%** of the parameters, and achieving a **21.06%** improvement in FPS.

## 2 RELATED WORK

### 2.1 3D OCCUPANCY PREDICTION

The 3D occupancy prediction task comprises two sub-tasks: predicting the geometric occupancy status for each voxel in 3D space and assigning corresponding semantic labels. We can categorize mainstream 3D occupancy networks into two architectures: CNN architecture based on the LSS (Philion & Fidler, 2020; Gan et al., 2023; Cao & de Charette, 2022; Yu et al., 2023; Mei et al., 2023; Ming et al., 2024; Hou et al., 2024) method and Transformer architecture based on the BEVFormer (Li et al., 2022; 2023a; Huang et al., 2023; Wei et al., 2023; Jiang et al., 2023; Wang et al., 2023; Liu et al., 2023) method. Due to the deployment advantages of CNN models, this paper focuses on CNN-based 3D occupancy networks. MonoScene (Cao & de Charette, 2022) is a pioneering work that utilizes a CNN framework to extract 2D features, which it then transforms into 3D representations. BEVDet-Occ (Huang & Huang, 2022) utilizes the LSS method to convert image features into BEV (Bird's Eye View) features and employs BEV pooling techniques to accelerate model inference. FlashOcc (Yu et al., 2023) replaces 3D convolutions in BEVDet-Occ with 2D convolutions and occupancy logits derived from 3D convolutions with channel-to-height transformations of BEV-level features obtained through 2D convolutions. SGN (Mei et al., 2023) adopts a dense-sparse-dense design and proposes hybrid guidance and efficient voxel aggregation to enhance intra-class feature separation and accelerate the convergence of semantic diffusion. InverseMatrixVT3D (Ming et al., 2024) introduces a new method based on projection matrices to construct local 3D feature volumes and global BEV features. Despite achieving impressive results, these CNN-based methods rely on powerful hardware with substantial computational and memory resources, which are impractical for edge devices. How to develop 3D occupancy prediction networks for resource-constrained devices remains underexplored. Our goal is to address this research gap.

### 2.2 BINARIZED NEURAL NETWORK

BNN (Hubara et al., 2016; Xia et al., 2022; Cai et al., 2024; Li et al., 2023c; Qin et al., 2024; Liu et al., 2020; 2018; Rastegari et al., 2016; Chen et al., 2021; Qin et al., 2020) represents the most extreme form of model quantization, quantizing weights and activations to just 1 bit. Due to its significant effectiveness in memory and computational compression, BNN (Hubara et al., 2016) finds wide application in both high-level vision and low-level vision. For instance, Xia et al. (Xia et al., 2022) designed a binarized convolutional unit, BBCU, for tasks such as image super-resolution, denoising, and reducing artifacts from JPEG compression. Cai et al. (Cai et al., 2024) devised a binarized convolutional unit, BiSR-Conv, capable of adjusting the density and distribution of representations for hyperspectral image (HSI) recovery. However, the potential of BNN in 3D occupancy tasks remains unexplored. Hence, this paper explores binarized 3D occupancy networks, aiming to maintain high performance while minimizing computational and parameter overhead.

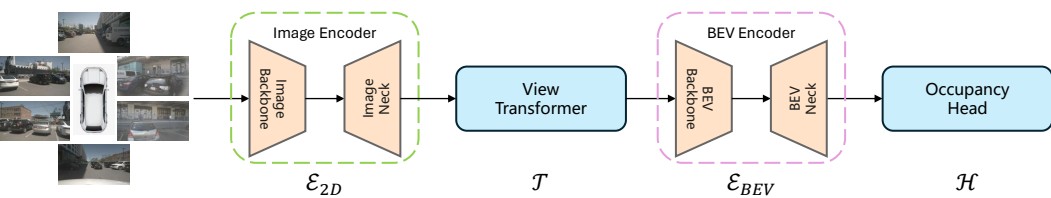

Figure 2: CNN-based 3D Occupancy Network

# 3 METHOD

## 3.1 BASE MODEL

The full-precision models to be binarized should be lightweight and easy to deploy on edge devices. However, prior 3D occupancy network models based on CNNs (He et al., 2016) or Transformers (Dosovitskiy et al., 2020; Liu et al., 2021) have high computational complexity or large model sizes. Some of these works utilize complex operations such as deformable attention, which are challenging to binarize and deploy on edge devices. Therefore, we redesign a simple, lightweight, and deployable baseline model without using complex computational operations.

BEVDet-Occ (Huang et al., 2021) and FlashOcc (Yu et al., 2023) demonstrate outstanding performance in 3D occupancy prediction tasks using only lightweight CNN architectures. Inspired by these works, we adopt the network structure shown in Figure 2 as our full-precision baseline model. It consists of an image encoder $\mathcal{E}_{2D}$, a view transformer module $\mathcal{T}$, a BEV encoder $\mathcal{E}_{BEV}$, and an occupancy head $\mathcal{H}$. The occupancy prediction network is composed of these modules concatenated sequentially. Assuming the input images are $\mathbf{I} \in \mathbb{R}^{N_{view} \times 3 \times H \times W}$, the occupancy prediction output $\mathbf{O} \in \mathbb{R}^{X \times Y \times Z}$ can be formulated as

$$\mathbf{O} = \mathcal{H}(\mathcal{E}_{BEV}(\mathcal{T}(\mathcal{E}_{2D}(\mathbf{I})))) \tag{1}$$

where $H$ and $W$ represent the height and width of the input images, and $X$, $Y$, and $Z$ denote the length, width, and height of the 3D space, respectively, $N_{view}$ represents the number of multi-view cameras. Please refer to the supplementary materials for a more detailed description of the base model.

## 3.2 BINARIZED DEEP CONVOLUTION

Due to its outstanding performance and lightweight architecture, FlashOcc (Yu et al., 2023) serves as the full-precision baseline model for the binarized model. Its performance reaches 37.84 mIoU, which sets the upper performance bound for the binarized models.

Empirical evidence in full-precision models has shown that increasing network depth improves performance. Due to the characteristics of binary networks, it is possible to maintain significantly low computational and memory usage even when increasing the model depth. However, in previous research, Xia et al. (Xia et al., 2022) observed that increasing the number of binarized convolutional layers within the binarized convolutional unit leads to a significant decrease in binarized model performance, the performance degradation issue with the increase in binarized convolutional layer depth within each unit restricts the further application of the binarized model. To address this issue, we propose the Binarized Deep Convolution (BDC) unit, which aims to enhance the binarized model performance by deepening the layers of the binarized convolution unit rather than reducing performance.

Cai et al. (Cai et al., 2024) proposed the binarized convolution unit BiSR-Conv, which can adjust the density and enable effective binarization of convolutional layers. We utilize BiSR-Conv to binarize FlashOcc (Yu et al., 2023), forming our initial version of **BDC-V0**, with its structure shown in Figure 3 (a). Please refer to the supplementary materials for a more detailed description of the BDC-V0. The model achieves a performance of **34.51 mIoU**.

**Theorem 1** (proven in the supplementary material). *In the process of backpropagation, we denote the expected value of the element-wise absolute gradient error of the parameters $\boldsymbol{w}$ in the $l$-th bina-*

*rized convolutional layer as $\mathbb{E}[\Delta \frac{\partial L}{\partial w_{mn}^{(l)}}]$. The specific expression is as follows.*

$$\mathbb{E}[\Delta \frac{\partial L}{\partial w_{mn}^{(l)}}] \leq 0.5354 \cdot (\sum_i \sum_j \sum_{m'=-(k//2)}^{k//2} \sum_{n'=-(k//2)}^{k//2} \mathbb{E}[|\frac{\partial \sigma(y_{(i+m')(j+n')}^{(l)})}{\partial y_{ij}^{(l)}} \cdot w_{m'n'}^{(l+1)} \cdot \frac{\partial L}{\partial y_{ij}^{(l+1)}}|]) \tag{2}$$

*where $k$ is the binarized convolution kernel size, $\frac{\partial \sigma(y_{(i+m')(j+n')}^{(l)})}{\partial y_{ij}^{(l)}}$ is the derivative of the activation function $\sigma(\cdot)$, $w_{m'n'}^{(l+1)}$ represents the weights of the binarized convolutional kernel in the next layer, and $\frac{\partial L}{\partial y_{ij}^{(l+1)}}$ is the element-wise gradient in the next layer.*

Based on Theorem 1, using a $3 \times 3$ convolutional kernel for binarized convolution leads to more binarization errors than a $1 \times 1$ kernel. Additionally, the model necessitates the presence of the first $3 \times 3$ binarized convolutional layer to maintain its capability for extracting local features. Therefore, building upon the binarized convolution unit BDC-V0, we introduce a $1 \times 1$ binarized convolutional layer after the $3 \times 3$ binarized convolution and before the residual connection, proposing **BDC-V1** as shown in Figure 3(b). By deepening the binarized convolution unit, BDC-V1 enhances its feature extraction capability while effectively balancing the trade-off introduced by binarization errors, achieving a performance of **36.29 mIoU**.

We seek to improve model performance by increasing the model's parameter count. Consequently, we added several $1 \times 1$ binary convolution layers to BDC-V1, resulting in the new model designated as **BDC-V2**. The structure of BDC-V2 is shown in Figure 3(c). We define the added multi-layer binarized convolution as MulBiconv$_N$, comprising $N$ RPReLU activations and $1 \times 1$ binarized convolutional layers, which can be expressed as

$$\text{MulBiconv}_N(\cdot) = \text{Repeat}_N(\text{Biconv1} \times 1(\text{RPReLU}(\cdot))) \tag{3}$$

where $\text{Repeat}_N(f)$ denotes repeating $N$ times operation $f$.

When $N = 1$, the performance drops to 35.88 mIoU; $N = 2$, it drops further to **35.43 mIoU**. We observe a decreasing trend in network performance as the number of $1 \times 1$ binarized convolutional layers increases. It occurs as the accumulated binarization errors increase with the addition of more binarized convolutional layers within the unit. The negative impact of binarization errors on the performance of binary models surpasses the positive effects of increased parameters, resulting in a decline in model performance.

### 3.3 PER-CHANNEL REFINEMENT BRANCH

**Theorem 2** (proven in the supplementary material). *Compared to full-precision convolutional layers, binarized convolutional layers exhibit disadvantages in capturing the scale variations across multiple channels of the feature maps. The specific expression is as follows.*

$$\sup_{X, \phi_{c_1}, \phi_{c_2}} |S_{\hat{y}^{c_1}} - S_{\hat{y}^{c_2}}| < \sup_{X, \phi_{c_1}, \phi_{c_2}} |S_{y^{c_1}} - S_{y^{c_2}}| \tag{4}$$

*Let $X \in \mathbb{R}^{C \times H \times W}$ represent the input feature maps, and let $\phi_c$ denote the full-precision convolution kernel of the c-th channel, which satisfies $avg(|\phi_c|) < max(|\phi_c|)$. The term $S$. refers to the scale of the feature map, defined as the normalized $\ell_1$-norm. Furthermore, $y$ and $\hat{y}$ represent the output feature map for a specific channel obtained from $\phi_c$ and its binarized version, respectively.*

In BNNs, all weights in each convolutional kernel share a unified scaling factor, with only the polarity varying. The cross-channel amplitude-frequency perception capability of full-precision convolution kernels degrades to a mere frequency response in binarized convolution. Based on Theorem 2, this characteristic of binary convolution hinders its ability to effectively integrate the attention of the input feature map across channels, leading to a suboptimal representation of inter-channel importance in the output feature maps. However, constructing robust inter-channel importance is essential for classification tasks (Hu et al., 2018) and is equally critical for occupancy prediction tasks, which focus on the classification of 3D samples.

Based on the above considerations, we propose the per-channel refinement branch, which forms the foundation of **BDC-V3**. The structure of the per-channel refinement branch is illustrated in

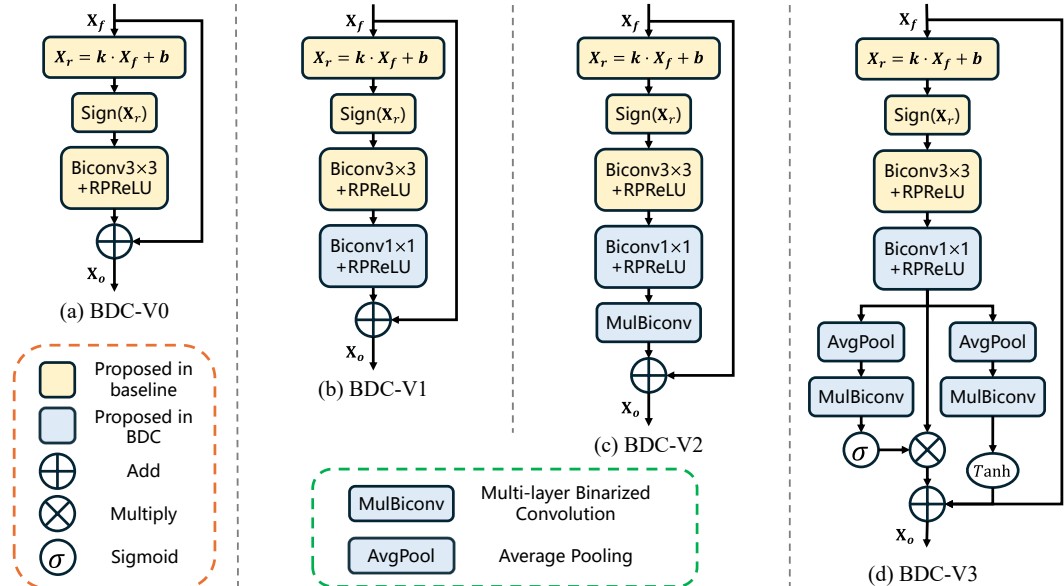

Figure 3: The illustration of the improvement process of our BDC.

Figure 3(d). First, the output of the first $1 \times 1$ binarized convolution, $\mathbf{X}_1$, served as the input for the per-channel refinement branch. The first-order and zero-order coefficients, designed to recover channel-wise scaling properties, are obtained through a dual-path structure comprising global average pooling (AvgPool), multi-layer binarized convolution (MulBiconv), and activation functions of Sigmoid and Tanh for each respective path. The branch output $\mathbf{Y}_1$ is formally expressed as

$$\mathbf{Y}_1 = \text{Sigmoid}(\text{MulBiconv}_N^A(\text{AvgPool}(\mathbf{X}_1))) \odot \mathbf{X}_1 + \text{Tanh}(\text{MulBiconv}_N^B(\text{AvgPool}(\mathbf{X}_1))) \quad (5)$$

where $\odot$ denotes element-wise multiplication. Through the proposed per-channel refinement branch, the newly introduced binarized convolutional layers reconstruct and enhance the cross-channel importance of the feature maps, enabling BDC-V3 to emulate the cross-channel feature extraction capability of full-precision convolution at first-order level. Additionally, from the perspective of Theorem 1, modeling the channel importance of feature maps through a first-order approximation enables the binarized model to focus more on channels less affected by binarization errors, thereby enhancing its perceptual capability.

When $N = 2$, the performance increased to **37.39 mIoU**, approaching the upper bound of 37.84 mIoU offered by the full-precision baseline model. We chose BDC-V3 with $N = 2$ as the final binarized convolutional unit, named **BDC**.

## 3.4 BINARIZED CONVOLUTION MODULE

Cai et al. (Cai et al., 2024) demonstrated the necessity of maintaining consistency in input and output dimensions for binarized convolutional layers to ensure the propagation of full-precision residual information. Consequently, specialized design considerations are necessary for each binarized convolution module. We can decompose the CNN-based occupancy network into four types of convolution modules:

(1) Basic convolution module: Input $\mathbf{X} \in \mathbb{R}^{C \times H \times W}$, output $\mathbf{Y} \in \mathbb{R}^{C \times H \times W}$;

(2) Down-sampling convolution module: Input $\mathbf{X} \in \mathbb{R}^{C \times H \times W}$, output $\mathbf{Y} \in \mathbb{R}^{2C \times \frac{H}{2} \times \frac{W}{2}}$;

(3) Up-sampling convolution module: Input $\mathbf{X} \in \mathbb{R}^{C \times H \times W}$, output $\mathbf{Y} \in \mathbb{R}^{C \times 2H \times 2W}$;

(4) Channel reduction convolution module: Input $\mathbf{X} \in \mathbb{R}^{C \times H \times W}$, output $\mathbf{Y} \in \mathbb{R}^{\frac{C}{2} \times H \times W}$;

We adopt a binarized design approach for these four convolution modules, leveraging methodologies from previous works (Liu et al., 2020; Xia et al., 2022; Cai et al., 2024), as illustrated in Figure 4. Figure 4 (a) illustrates the basic convolutional module, preserving both the size and the number of channels in the input feature map. Figure 4 (b) depicts the downsample convolution module,

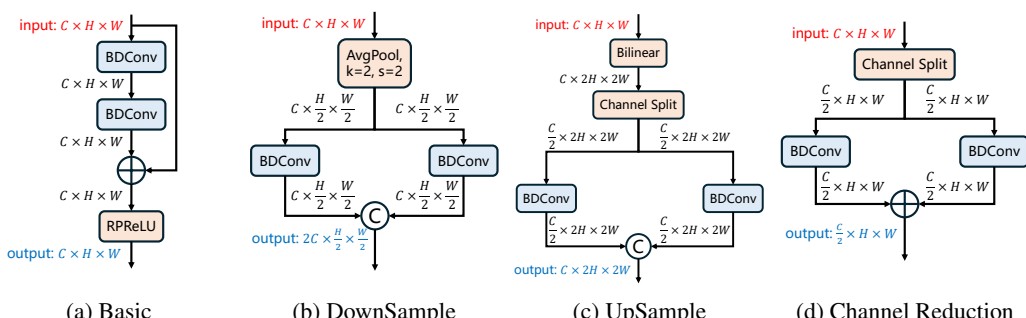

(a) Basic             (b) DownSample           (c) UpSample         (d) Channel Reduction

Figure 4: The illustration of binarized convolution module based on BDC.

reducing the size of the input feature map by half and doubling the number of channels. Figure 4 (c) showcases the upsample convolution module, doubling the size of the input feature map while preserving the number of channels. Finally, Figure 4 (d) presents the channel reduction convolution module, maintaining the size of the input feature map while halving the number of channels.

## 4 EXPERIMENT

### 4.1 EXPERIMENTAL SETTINGS

**Datasets.** We use the Occ3D-nuScenes dataset (Tian et al., 2023), which comprises 28,130 samples for training and 6,019 samples for validation.

**Evaluation Metrics.** We evaluate the Occ3D-nuScenes' validation set using the mean Intersection over Union (mIoU) metric. Similar to (Hubara et al., 2016), we compute the operations per second of BNN ($\text{OPs}^b$) as $\text{OPs}^b = \text{OPs}^f / 64$ to measure the computational complexity, where $\text{OPs}^f$ represents FLOPS. To calculate the parameters of BNN, use the formula $\text{Parms}^b = \text{Parms}^f / 32$, where the superscript $b$ and $f$ refer to the binarized and full-precision models, respectively. To compute the total operations and parameters, we sum OPs as $\text{OPs}^b + \text{OPs}^f$ and Params as $\text{Params}^b + \text{Params}^f$.

**Implementation Details.** For 3D occupancy prediction tasks, we employ FlashOcc (Yu et al., 2023) as the baseline network. We utilized ResNet50 (He et al., 2016) as the image backbone, with an input size of $256 \times 704$. Default learning rate $1 \times 10^{-4}$, AdamW (Loshchilov & Hutter, 2017) optimizer, and weight decay of $1 \times 10^{-2}$ were utilized. The training lasted approximately 29 hours, utilizing 24 epochs on two NVIDIA 3090 GPUs, with a batch size of 2 per GPU. Data augmentation strategies for the Occ3D-nuScenes dataset remained consistent with those of FlashOcc (Yu et al., 2023). Previous works, such as FlashOcc and BEVDet-Occ (Huang & Huang, 2022), have demonstrated the effectiveness of camera visibility masks during training. Therefore, we also employ camera visibility masks to enhance performance. Following the settings of FlashOcc, we employ the pre-trained model from BEVDet (Huang et al., 2021) for 3D object detection tasks as our pre-training model.

### 4.2 MAIN RESULTS

To ensure performance, we refrain from binarizing the image backbone in the image encoder. This component contains pre-trained weights from image classification tasks, effectively facilitating model convergence and incorporating prior semantic information from images. We binarize the BEV encoder and occupancy head as the **base** version (**-B**) for all binarized models. We further binarize the image neck in the image encoder to obtain the **tiny** version (**-T**) based on the base version.

Table 1 presents the evaluation results of our method BDC on the validation set of Occ3D-nuScenes. To validate the effectiveness of our proposed method BDC, we compare it with other state-of-the-art binarized models, including ReActNet (Liu et al., 2020), PokeBNN (Zhang et al., 2022), AdaBin (Tu et al., 2022), BBCU (Xia et al., 2022), BiMatting (Li et al., 2023c), and BiSRNet (Cai et al., 2024). We also compare it with full-precision occupancy prediction networks based on CNN architectures, including BEVDet-Occ (Huang et al., 2021) and FlashOcc (Yu et al., 2023), where FlashOcc serves as the baseline network for all binarized models and represents the theoretical upper limit of binarized model performance.

Table 1: **Occupancy Prediction performance (mIoU↑) on the Occ3D-nuScenes datasets.** Best and second best performance among BNNs are in red and blue colors, respectively.

| Methods | Params(M) | OPs(G) | others | barrier | bicycle | bus | car | const. veh. | motorcycle | pedestrian | traffic cone | trailer | truck | drive. suf. | other flat | sidewalk | terrain | manmade | vegetation | mIoU |
|---|---|---|---|---|---|---|---|---|---|---|---|---|---|---|---|---|---|---|---|---|
| *CNN-based (32 bit)* | | | | | | | | | | | | | | | | | | | | |
| BEVDet-Occ | 29.02 | 241.76 | 8.22 | 44.21 | 10.34 | 42.08 | 49.63 | 23.37 | 17.41 | 21.49 | 19.70 | 31.33 | 37.09 | 80.13 | 37.37 | 50.41 | 54.29 | 45.56 | 39.59 | 36.01 |
| FlashOcc | 44.74 | 248.57 | 9.08 | 46.32 | 17.71 | 42.70 | 50.64 | 23.72 | 20.13 | 22.34 | 24.09 | 30.26 | 37.39 | 81.68 | 40.13 | 52.34 | 56.46 | 47.69 | 40.60 | 37.84 |
| *BNN-based (1 bit)* | | | | | | | | | | | | | | | | | | | | |
| ReaActNet-T | 26.80 | 129.74 | 7.55 | 38.87 | 16.64 | 35.78 | 44.27 | 20.34 | 15.53 | 16.16 | 18.70 | 24.42 | 33.59 | 73.64 | 29.05 | 39.80 | 41.27 | 39.31 | 34.00 | 31.29 |
| ReaActNet-B | 28.17 | 133.89 | 8.62 | 40.92 | 15.94 | 37.45 | 47.23 | 18.57 | 17.47 | 18.91 | 21.52 | 23.14 | 33.13 | 77.20 | 34.58 | 45.48 | 48.31 | 42.95 | 35.06 | 33.32 |
| PokeBNN-T | 26.81 | 129.84 | 6.64 | 42.26 | 21.80 | 36.29 | 47.78 | 22.08 | 21.33 | 20.90 | 21.69 | 26.09 | 34.92 | 78.82 | 37.75 | 46.79 | 49.50 | 44.40 | 38.64 | 35.16 |
| AdaBin-T | 26.78 | 129.78 | 8.21 | 40.59 | 17.12 | 37.02 | 46.92 | 21.18 | 18.67 | 19.40 | 19.79 | 24.56 | 34.47 | 76.62 | 19.77 | 44.75 | 48.22 | 43.87 | 37.57 | 32.87 |
| BBCU-T | 26.79 | 129.69 | 6.24 | 38.16 | 14.33 | 31.95 | 43.18 | 20.57 | 16.50 | 17.39 | 13.45 | 22.26 | 32.51 | 75.69 | 32.97 | 42.46 | 48.50 | 41.68 | 35.75 | 31.39 |
| BBCU-B | 28.16 | 133.84 | 7.61 | 41.14 | 13.64 | 35.54 | 46.55 | 20.86 | 17.44 | 19.87 | 17.58 | 24.24 | 33.94 | 76.19 | 34.05 | 44.61 | 48.08 | 42.67 | 35.28 | 32.27 |
| BiMatting-T | 26.82 | 129.95 | 5.96 | 38.17 | 15.27 | 35.85 | 44.11 | 19.35 | 14.38 | 18.98 | 15.84 | 23.22 | 31.16 | 73.97 | 30.51 | 35.42 | 40.90 | 41.65 | 35.05 | 30.58 |
| BiMatting-B | 28.17 | 134.05 | 6.80 | 38.65 | 17.99 | 33.02 | 43.80 | 19.91 | 18.29 | 18.67 | 19.82 | 21.83 | 32.09 | 72.99 | 32.44 | 41.23 | 43.64 | 36.24 | 35.07 | 31.32 |
| BiSRNet-T | 26.79 | 129.70 | 8.38 | 41.06 | 16.76 | 33.94 | 46.11 | 18.96 | 19.10 | 17.90 | 16.94 | 23.70 | 35.14 | 76.86 | 35.68 | 46.77 | 50.39 | 41.41 | 34.78 | 33.17 |
| BiSRNet-B | 28.16 | 133.85 | 9.27 | 41.94 | 19.53 | 37.33 | 47.48 | 20.83 | 19.17 | 20.08 | 20.21 | 25.36 | 33.99 | 77.42 | 35.78 | 47.35 | 50.58 | 43.24 | 37.20 | 34.51 |
| **BDC-T (Ours)** | 26.83 | 129.90 | 10.16 | 44.38 | 18.53 | 41.40 | 49.87 | 23.12 | 20.94 | 22.33 | 23.29 | 29.93 | 36.19 | 81.14 | 39.37 | 51.43 | 55.25 | 47.37 | 40.87 | 37.39 |
| **BDC-B (Ours)** | 28.22 | 134.50 | 9.57 | 44.80 | 20.45 | 40.21 | 49.96 | 23.72 | 21.48 | 22.58 | 24.47 | 27.40 | 36.48 | 80.22 | 38.34 | 50.12 | 54.74 | 47.19 | 40.04 | 37.16 |

Table 2: **3D Object Detection performance (mAP↑, NDS↑) on the nuScenes `val` set.** Best performance among BNNs are in **bold**.

| Methods | Params(M) | OPs(G) | **mAP↑** | **NDS↑** | mATE↓ | mASE↓ | mAOE↓ | mAVE↓ | mAAE↓ |
|---|---|---|---|---|---|---|---|---|---|
| *CNN-based (32 bit)* | | | | | | | | | |
| BEVDet | 44.25 | 148.77 | 0.3836 | 0.4995 | 0.5815 | 0.2790 | 0.4750 | 0.3807 | 0.2067 |
| *BNN-based (1 bit)* | | | | | | | | | |
| ReactNet-T | 26.53 | 101.30 | 0.3222 | 0.4358 | 0.6609 | 0.3057 | 0.6298 | 0.4468 | 0.2100 |
| BBCU-T | 26.51 | 101.24 | 0.3166 | 0.4046 | 0.6697 | 0.3137 | 0.7822 | 0.5461 | 0.2255 |
| BiMatting-T | 26.55 | 101.41 | 0.3356 | 0.4428 | 0.6358 | 0.2968 | 0.6527 | 0.4485 | 0.2159 |
| BiSRNet-T | 26.52 | 101.25 | 0.3431 | 0.4519 | 0.6633 | 0.2940 | 0.5777 | 0.4550 | 0.2061 |
| BDC-T | 26.56 | 101.36 | **0.3648** | **0.4742** | **0.6291** | **0.2822** | **0.5250** | **0.4460** | **0.1994** |

Table 1 presents performance metrics (mIoU), parameter counts, and the number of operations for different methods. Compared to other binarized methods, our BDC-T and BDC-B achieve the best or second-best results across almost all binarized models. Specifically, BDC significantly improves performance without increasing parameter count or computational complex-

Table 3: **Computational efficiency.** FPS and Run time (ms) for 32-bit and 1-bit of FlashOcc and BDC-T

| Methods | 32 bit | 1 bit | total time | FPS |
|---|---|---|---|---|
| FlashOcc | 160.77 | 0 | 160.77 | 6.22 |
| BDC-T | 130.93 | 1.88 | 132.81 | 7.53 |

ity. Compared to the previous SOTA method, BiSRNet-B, our BDC-T demonstrates superior performance in mIoU, exceeding it by 2.88 mIoU (**+8.35%**) while saving 2.95% of operations and 4.72% of parameters. Moreover, BDC-T achieves competitive results compared to the full-precision model FlashOcc, using only **52.26%** of operations and **59.97%** of parameters, with a minimal performance loss of -0.45 mIoU (-1.19%) due to binarization errors. Both BBCU and BiSRNet exhibit performance degradation issues when binarizing additional modules. Compared to BDC-B, BDC-T performs slightly better when binarizing image neck modules. It demonstrates the robustness of BDC to the binarized modules. In Table 3, we compare the wall-clock time computational efficiency, showing that our model achieves a **21.06%** improvement in FPS.

To validate the generalizability of the proposed BDC, we also conduct experiments on 3D object detection tasks using the nuScenes (Caesar et al., 2020) dataset. Table 2 presents performance metrics for the 3D object detection task in nuScenes, where our approach, BDC, continues to demonstrate superior performance in both mAP and NDS.

Table 4: **Break-down ablation.** Figure 3 illustrates the structure of various versions of the BDC.

| Methods | mIoU | OPs (G) | Params (M) |
|---|---|---|---|
| BDC-V0 | 34.51 | 133.85 | 28.16 |
| BDC-V1 | 36.29 | 133.93 | 28.17 |
| BDC-V2 | 35.43 | 134.10 | 28.19 |
| BDC-V3 | **37.16** | 134.50 | 28.22 |

Table 5: **Kernel size ablation.** $A \to B$ represents the concatenation structure of $A \times A$ binarized convolution followed by $B \times B$ binarized convolution.

| Kernel | mIoU | OPs (G) | Params (M) |
|---|---|---|---|
| $3 \to 1$ | **36.29** | 133.93 | 28.17 |
| $3 \to 3$ | 33.01 | 133.93 | 28.17 |
| $1 \to 1$ | 35.32 | 133.93 | 28.17 |
| $3 \to 3 \to 1$ | 33.37 | 134.02 | 28.18 |

### 4.3 ABLATION STUDY

In all ablation studies, the binarization settings are configured as the **base** version (**-B**) for all models as described in Table 1.

**Multi-layer Binarized Convolution (MulBiconv) Ablation.** To explore the impact of the number of binarized convolutional layers in MulBiconv on the model's performance, we binarize FlashOcc using both BDC-V2 and BDC-V3 while varying the number of binarized convolutional layers in MulBiconv ($N = 0, 1, 2, 3, 4$).

The results are illustrated in Figure 5. When $N = 0$, the structure of BDC-V2 is identical to that of BDC-V1. BDC-V3 contains no learnable parameters with the per-channel refinement branch. As $N$ increases, we observe a gradual decline followed by fluctuations in the performance of BDC-V2. In contrast, BDC-V3 initially shows performance improvement, fol-

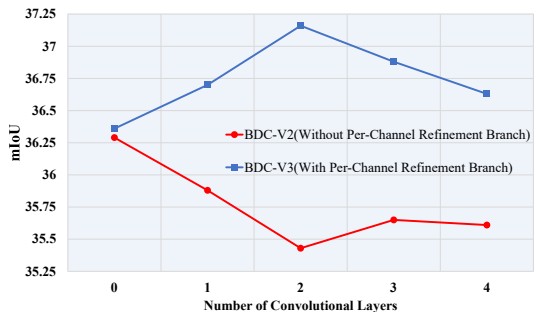

Figure 5: Ablation study of multi-layer binarized convolution (MulBiconv)

lowed by decreases as $N$ increases. When MulBiconv selects $N = 2$, BDC-V3 achieves the best performance, reaching 37.16 mIoU. The optimal trade-off occurs when the performance gain from reducing model parameters outweighs the performance degradation caused by binarization errors.

**Break-down Ablation.** We binarize FlashOcc using four variants of BDC, where BDC-v0 is equivalent to the binarized method BiSRNet. Additionally, BDC-V2 and BDC-V3 utilize the multi-layer binarized convolution (MulBiconv), and we set $N = 2$.

The results are presented in Table 4, from which we can draw the following conclusions: (1) Compared to BDC-V0, BDC-V1 achieves a significant gain of **1.78 mIoU (+5.16%)** by adding only one $1 \times 1$ binarized convolution layer. Extra binarized convolution layers result in negligible changes to full model parameters and computational complexity. (2) By adding MulBiconv to each binarized convolution unit in BDC-V1 (i.e., BDC-V2), we observe a substantial decrease in performance, along with slight increases in parameters and computational complexity. (3) Compared to BDC-V2, BDC-V3 exhibits a significant performance improvement of **1.73 mIoU**. Additionally, BDC-V3 gains an extra **0.87 mIoU** over BDC-V1. Placing additional binarized convolutional layers within the per-channel refinement branch effectively enhances model performance.

**Kernel Size Ablation.** To validate whether $3 \times 3$ binarized convolutions incur more binarization errors than $1 \times 1$ ones, potentially leading to performance degradation, we apply BDC-V1 and BDC-V2 ($N = 1$) to FlashOcc. We present the results in Table 5. For BDC-V1, replacing the $1 \times 1$ binarized convolution with consecutive $3 \times 3$ binarized convolutions led to a decrease in performance from 36.29 mIoU to 33.01 mIoU.

Additionally, we validate the necessity of using a $3 \times 3$ binarized convolution as the first convolution layer. If replaced with a $1 \times 1$ binarized convolution, the receptive field of the binarized convolution unit becomes limited, preventing the establishment of connections with neighboring pixel features, resulting in a decrease in performance from 36.29 mIoU to 35.32 mIoU. Experiments conducted on

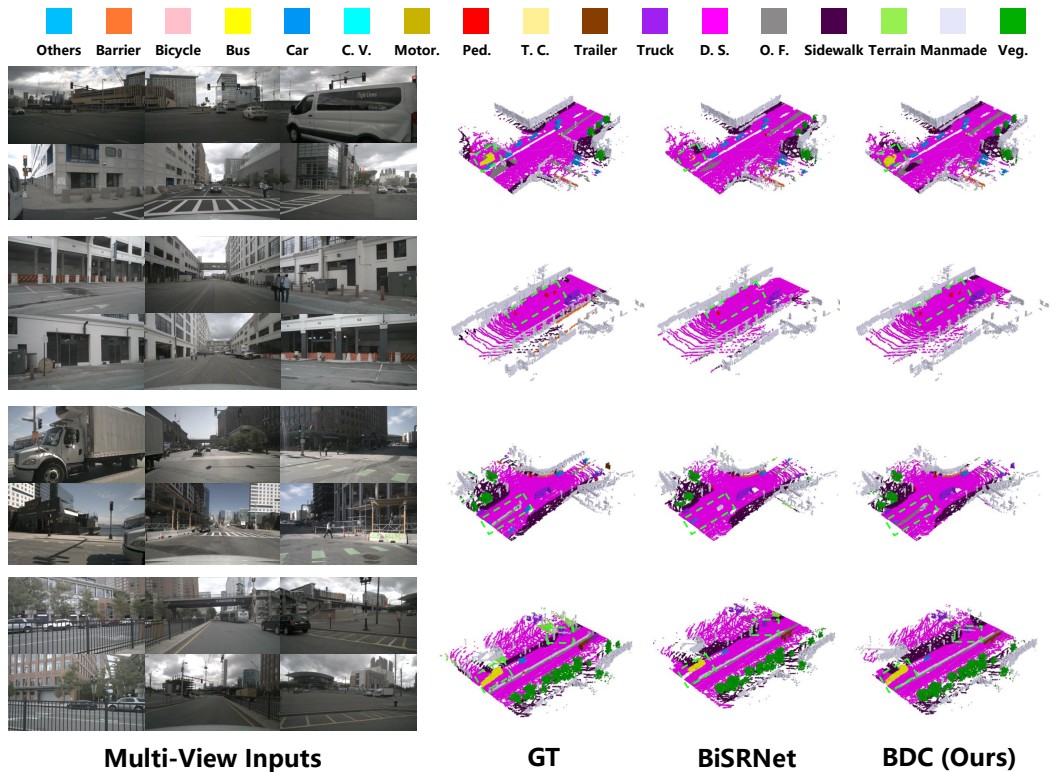

Figure 6: Visualization rensults on Occ3D-nuScenes validation set

BDC-V2 ($N = 1$) also support the conclusion that consecutive $3 \times 3$ binarized convolutions lead to binarization errors and affect binarized model performance.

## 4.4 VISUALIZATION

We also present some qualitative results on the Occ3D-nuScenes' validation set. As illustrated in Figure 6, BDC exhibits comprehensive predictions about the bus in the first and last rows. In the second row, BDC successfully identifies all pedestrians, whereas BiSRNet overlooks some pedestrians in the scene. Moreover, in the third row, BDC provides accurate predictions about curbs, whereas BiSRNet misclassifies them as drivable surfaces, potentially posing safety concerns. Additionally, in the fourth row, BDC accurately reconstructs traffic lights in the scene, showcasing its robust capability in scene perception.

## 5 CONCLUSION

This paper introduces a binarized deep convolution (BDC) unit for 3D occupancy networks, addressing the performance degradation caused by increasing the number of binarized convolutional layers. Our original theoretical analysis shows that $1 \times 1$ binarized convolution introduces minimal binarization errors, and binarized convolution is less effective than full-precision convolution in capturing cross-channel feature importance. Consequently, we restrict additional binarized convolution kernels to $1 \times 1$ in the BDC unit. Furthermore, we propose a per-channel refinement branch to overcome the limitations of binarized convolutional layers in assigning feature importance across channels. Extensive experiments validate that our method surpasses existing SOTA binarized convolution networks and closely approaches the performance of full-precision models while using only **52.26%** of the operations and **59.97%** of the parameters and achieving a **21.06%** improvement in FPS.

**Limitation.** We have not tested our method for performance in Transformer architectures, which may limit its broader application.

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

# A    APPENDIX

## A.1    MORE DETAILS ABOUT BASE MODEL

Base model consists of an image encoder $\mathcal{E}_{2D}$, a view transformer module $\mathcal{T}$, a BEV encoder $\mathcal{E}_{BEV}$, and an occupancy head $\mathcal{H}$. The occupancy prediction network is composed of these modules concatenated sequentially. Assuming the input images are $\mathbf{I} \in \mathbb{R}^{N_{view} \times 3 \times H \times W}$, the occupancy prediction output $\mathbf{O} \in \mathbb{R}^{X \times Y \times Z}$ can be formulated as

$$\mathbf{O} = \mathcal{H}(\mathcal{E}_{BEV}(\mathcal{T}(\mathcal{E}_{2D}(\mathbf{I})))) \tag{6}$$

where $H$ and $W$ represent the height and width of the input images, and $X$, $Y$, and $Z$ denote the length, width, and height of the 3D space, respectively, $N_{view}$ represents the number of multi-view cameras.

First, Multi-view images are sent to the image encoder $\mathcal{E}_{2D}$ to obtain 2D features $\mathbf{f}_{2D} \in \mathbb{R}^{N_{view} \times C_{2D} \times H_{2D} \times W_{2D}}$ and depth prediction $\mathbf{f}_{depth} \in \mathbb{R}^{N_{view} \times N_{depth} \times H_{2D} \times W_{2D}}$, where $C_{2D}, H_{2D}, W_{2D}$ denote the number of channels, height and width of 2D features, respectively. $N_{depth}$ represents the number of depth bins in the depth prediction.

Subsequently, the image features $\mathbf{f}_{2D}$ and depth prediction $\mathbf{f}_{depth}$ are passed through the visual transformation module $\mathcal{T}$, which transforms them into primary BEV features $\mathbf{f}_T \in \mathbb{R}^{C_{BEV} \times H_{BEV} \times W_{BEV}}$ using camera intrinsic and extrinsic projection matrices. Here, $C_{BEV}$ represents the number of channels of BEV features, while $H_{BEV}$ and $W_{BEV}$ represent the length and width of the BEV space, respectively. Since the voxel distribution obtained from the depth map through projection matrices is sparse, the representation capability of primary BEV features may be insufficient. To this end, $\mathbf{f}_T$ is passed through the BEV encoder $\mathcal{E}_{BEV3D}$ to obtain fine BEV features $\mathbf{f}_{BEV} \in \mathbb{R}^{C_{BEV} \times H_{BEV} \times W_{BEV}}$ for further refinement.

Finally, the semantic prediction output logits $\mathbf{O}_{logits} \in \mathbb{R}^{N_{class} \times X \times Y \times Z}$ come from the BEV features $\mathbf{f}_{BEV}$ processed through the occupancy prediction head $\mathcal{H}$, where $N_{class}$ is the number of semantic classes in the dataset. By taking the index corresponding to the maximum value of the logits, we can obtain the final occupancy prediction output $\mathbf{O}$.

## A.2    MORE DETAILS ABOUT BDC-V0

We define BDC-V0 following the method proposed in BiSRNet Cai et al. (2024). Both full-precision image features and Bird's Eye View (BEV) features, represented as $\mathbf{X}_f \in \mathbb{R}^{C \times H \times W}$, serve as input for the full-precision activations.

Figure 7: The schematic diagram of binarized convolution Rastegari et al. (2016).

In 3D occupancy networks, features transform from dense 2D space to sparse 3D space and then back to dense 3D space, causing significant differences in feature distribution. Each module has distinct densities and distributions.

To address the problem of significant differences in feature distribution, we follow the approach of BiSRNet, employing channel-wise feature redistribution:

$$\mathbf{X}_r = k \cdot \mathbf{X}_f + b \tag{7}$$

Here, $\mathbf{X}_r \in \mathbb{R}^{C \times H \times W}$ represents the activations after channel-wise feature redistribution, and $k, b \in \mathbb{R}^C$ are learnable parameters. $k$ represents the learnable density of redistribution, while $b$ represents the learnable bias of redistribution.

Next, $\mathbf{X}_r$ is passed through the Sign function to binarize it, yielding 1-bit binarized activations $\mathbf{X}_b \in \mathbb{R}^{C \times H \times W}$, as follows:

$$x_b = \text{Sign}(x_r) = \begin{cases} +1, & \text{if } x_r > 0 \\ -1, & \text{if } x_r \leq 0 \end{cases} \tag{8}$$

where $x_r \in \mathbf{X}_r$, $x_b \in \mathbf{X}_b$.

Since the Sign function is not differentiable, approximation functions are required to ensure successful backpropagation. Common approximation functions include piecewise linear function $\text{Clip}(\cdot)$, piecewise quadratic function $\text{Quad}(\cdot)$, and hyperbolic tangent function $\text{Tanh}(\cdot)$. We use the hyperbolic tangent function as the approximation function, defined as:

$$x_b = \text{Tanh}(\alpha x_r) = \frac{e^{\alpha x_r} - e^{-\alpha x_r}}{e^{\alpha x_r} + e^{-\alpha x_r}} \tag{9}$$

The Tanh function ensures gradients exist even when weights and activations exceed 1, allowing parameter updates downstream during backpropagation.

In the binarized convolutional layer, the 32-bit precision weights $\mathbf{W}_f$ are binarized into 1-bit binarized weights $\mathbf{W}_b$ according to the following formula:

$$w_b = \mathbb{E}_{w_f \in \mathbf{W}_f}(|w_f|) \cdot \text{Sign}(w_f) \tag{10}$$

Here, $\mathbb{E}_{w_f \in \mathbf{W}_f}(|w_f|)$ represents the average absolute value of the full-precision weights, which serves as a scaling factor to reduce the discrepancy between the binarized weights $\mathbf{W}_b$ and the full-precision weights $\mathbf{W}_f$. Multiplying this value by $\text{Sign}(w_f) = \pm 1$ yields element-wise binarized weights $w_b$.

Subsequently, the binarized activation $\mathbf{X}_b$ is convolved with the binarized weights $\mathbf{W}_b$. Binarized convolution can be accomplished purely through logical operations. The schematic diagram of binarized convolution Rastegari et al. (2016) is illustrated in Figure 7, and the expression is as follows:

$$\mathbf{Y}_b = \text{Biconv}(\mathbf{X}_b, \mathbf{W}_b) = \text{BitCount}(\text{XNOR}(\mathbf{X}_b, \mathbf{W}_b)) \tag{11}$$

Here, $\mathbf{Y}_b$ is the output of binarized convolution, Biconv denotes the binarized convolution layer, and BitCount and XNOR represent the bit count and logical XOR operations, respectively. In BDC-V0, the convolutional kernel size is $3 \times 3$.

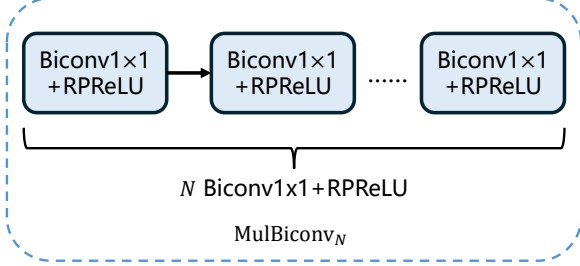

Figure 8: The struct of MulBiconv

For the activation function, we utilize RPReLU, whose expression is defined as follows:

$$\text{RPReLU}(y_i) = \begin{cases} y_i - \gamma_i + \zeta_i, & \text{if } y_i > \gamma_i \\ \beta_i \cdot (y_i - \gamma_i) + \zeta_i, & \text{if } y_i \leq \gamma_i \end{cases} \tag{12}$$

Here, $y_i \in \mathbb{R}$ represents the $i$-th element value of $\mathbf{Y}_b$, and $\beta_i$, $\gamma_i$, and $\zeta_i$ are learnable parameters for the $i$-th channel.

### A.3 MORE DETAILS ABOUT MULTIBICONV

The structure of MulBiconv, illustrated in Figure 8, is composed of multiple $1 \times 1$ binary convolution layers and RPReLU.

### A.4 PROOF OF THEOREM 1

**Theorem 1.** *In the process of backpropagation, we denote the expected value of the element-wise absolute gradient error of the parameters $\mathbf{w}$ in the $l$-th binarized convolutional layer as $\mathbb{E}[\Delta \frac{\partial L}{\partial w_{mn}^{(l)}}]$. The specific expression is as follows:*

$$\mathbb{E}[\Delta \frac{\partial L}{\partial w_{mn}^{(l)}}] \approx 0.5354 \cdot (\sum_i \sum_j \sum_{m'=-(k//2)}^{k//2} \sum_{n'=-(k//2)}^{k//2} \mathbb{E}[|\frac{\partial \sigma(y_{(i+m')(j+n')}^{(l)})}{\partial y_{ij}^{(l)}} \cdot w_{m'n'}^{(l+1)} \cdot \frac{\partial L}{\partial y_{ij}^{(l+1)}}|]) \tag{13}$$

*where $k$ is the binarized convolution kernel size, $\frac{\partial \sigma(y_{(i+m')(j+n')}^{(l)})}{\partial y_{ij}^{(l)}}$ is the derivative of the activation function $\sigma(\cdot)$, $w_{m'n'}^{(l+1)}$ represents the weights of the binarized convolutional kernel in the next layer, and $\frac{\partial L}{\partial y_{ij}^{(l+1)}}$ is the element-wise gradient in the next layer.*

**Proof.** We assume the element of the input of a binarized convolutional layer as $x_{ij}$, with a binarization error denoted as $\epsilon_{ij}$, the full-precision input before binarization as $\hat{x}_{ij}$, and the output of the binarized convolutional layer as $y_{ij}$. Thus, we have:

$$x_{ij} = \hat{x}_{ij} + \epsilon_{ij} \tag{14}$$

Since the full-precision input $\hat{x}_{ij}$ at the current layer is the output from the batch normalization layer in the previous layer, we can assume that the full-precision input $\hat{x}_{ij}$ follows a Gaussian distribution $\mathcal{N}(0, 1)$. Based on Equations equation 8 and equation 14, we can then derive the distribution of $\epsilon_{ij}$ as follows:

$$|\epsilon_{ij}| = |\hat{x}_{ij} - x_{ij}| = |\hat{x}_{ij} - \text{Sign}(\hat{x}_{ij})| = \begin{cases} |\hat{x}_{ij} - 1|, & \text{if } \hat{x}_{ij} > 0 \\ |\hat{x}_{ij} + 1|, & \text{if } \hat{x}_{ij} \leq 0 \end{cases} \tag{15}$$

Assuming the convolution kernel size $k$ is odd, for a $k \times k$ convolutional layer, the kernel weight $w_{mn}$, and the kernel bias is $b_{mn}$. The forward propagation equation is given by:

$$y_{ij} = \sum_{m=-(k//2)}^{k//2} \sum_{n=-(k//2)}^{k//2} (x_{(i+m)(j+n)} \cdot w_{mn} + b_{mn}) \tag{16}$$

Assuming that during backpropagation, the gradient at current layer $l$ is given by $\frac{\partial L}{\partial y_{ij}^{(l)}}$, we can use the chain rule to derive the gradient for a $k \times k$ convolutional layer as follows:

$$\frac{\partial L}{\partial w_{mn}^{(l)}} = \sum_i \sum_j x_{(i+m)(j+n)}^{(l)} \frac{\partial L}{\partial y_{ij}^{(l)}} = \sum_i \sum_j (\hat{x}_{(i+m)(j+n)}^{(l)} + \epsilon_{(i+m)(j+n)}^{(l)}) \cdot \frac{\partial L}{\partial y_{ij}^{(l)}} \tag{17}$$

Given that the output of the current layer $y_{ij}^{(l)}$ becomes the input of the next layer after passing through the activation function $\sigma(\cdot)$. Based on Equation equation 16, we can derive:

$$y_{ij}^{(l+1)} = \sum_{m'=-(k//2)}^{k//2} \sum_{n'=-(k//2)}^{k//2} \sigma(y_{(i+m')(j+n')}^{(l)}) \cdot w_{m'n'}^{(l+1)} + b_{m'n'}^{(l+1)} \tag{18}$$

We can obtain the gradient relationship between $\frac{\partial L}{\partial y_{ij}^{(l)}}$ and $\frac{\partial L}{\partial y_{ij}^{(l+1)}}$:

$$\frac{\partial L}{\partial y_{ij}^{(l)}} = \sum_{m'=-(k//2)}^{k//2} \sum_{n'=-(k//2)}^{k//2} \frac{\partial \sigma(y_{(i+m')(j+n')}^{(l)})}{\partial y_{ij}^{(l)}} \cdot w_{m'n'}^{(l+1)} \cdot \frac{\partial L}{\partial y_{ij}^{(l+1)}} \tag{19}$$

By substituting Equation equation 19 into Equation equation 17, we can obtain:

$$\frac{\partial L}{\partial w_{mn}^{(l)}} = \sum_i \sum_j \sum_{m'} \sum_{n'} (\hat{x}_{(i+m)(j+n)}^{(l)} + \epsilon_{(i+m)(j+n)}^{(l)}) \cdot \frac{\partial \sigma(y_{(i+m')(j+n')}^{(l)})}{\partial y_{ij}^{(l)}} \cdot w_{m'n'}^{(l+1)} \cdot \frac{\partial L}{\partial y_{ij}^{(l+1)}} \tag{20}$$

We can derive the additional gradient error $\Delta \frac{\partial L}{\partial w_{mn}^{(l)}}$ induced by the binarization error $\epsilon$ as follows:

$$\Delta \frac{\partial L}{\partial w_{mn}^{(l)}} := |\frac{\partial L}{\partial w_{mn}^{(l)}} - \frac{\partial L}{\partial w_{mn}^{(l)}}|_{\epsilon=0}|$$

$$= |\sum_i \sum_j \sum_{m'} \sum_{n'} \epsilon_{(i+m)(j+n)}^{(l)} \cdot \frac{\partial \sigma(y_{(i+m')(j+n')}^{(l)})}{\partial y_{ij}^{(l)}} \cdot w_{m'n'}^{(l+1)} \cdot \frac{\partial L}{\partial y_{ij}^{(l+1)}}| \tag{21}$$

$$\leq \sum_i \sum_j \sum_{m'} \sum_{n'} |\epsilon_{(i+m)(j+n)}^{(l)} \cdot \frac{\partial \sigma(y_{(i+m')(j+n')}^{(l)})}{\partial y_{ij}^{(l)}} \cdot w_{m'n'}^{(l+1)} \cdot \frac{\partial L}{\partial y_{ij}^{(l+1)}}|$$

By utilizing Equation equation 15, we can calculate the expected value of the absolute binarization error, denoted as $\mathbb{E}[|\epsilon_{ij}|]$:

$$\mathbb{E}[|\epsilon_{ij}|] = \int_0^\infty |\hat{x}_{ij} - 1| \frac{1}{\sqrt{2\pi}} e^{-\frac{\hat{x}_{ij}^2}{2}} \, d\hat{x}_{ij} + \int_{-\infty}^0 |\hat{x}_{ij} + 1| \frac{1}{\sqrt{2\pi}} e^{-\frac{\hat{x}_{ij}^2}{2}} \, d\hat{x}_{ij}$$

$$= 2(\int_0^1 \frac{1 - \hat{x}_{ij}}{\sqrt{2\pi}} e^{-\frac{\hat{x}_{ij}^2}{2}} \, d\hat{x}_{ij} - \int_1^\infty \frac{1 - \hat{x}_{ij}}{\sqrt{2\pi}} e^{-\frac{\hat{x}_{ij}^2}{2}} \, d\hat{x}_{ij}) \tag{22}$$

The Gaussian error function, often abbreviated as "$erf(x)$" is defined as follows:

$$erf(x) = \frac{2}{\sqrt{\pi}} \int_0^x e^{-t^2} \, dt \tag{23}$$

Based on the definition of the Gaussian error function and the use of the substitution rule, we can compute the integral as follows:

$$\int_0^x e^{-\frac{u^2}{2}} \, du \xupequal{u=\sqrt{2}t} \sqrt{2} \int_0^{\frac{x}{\sqrt{2}}} e^{-t^2} \, dt$$

$$= \frac{\sqrt{\pi}}{\sqrt{2}} \frac{2}{\sqrt{\pi}} \int_0^{\frac{x}{\sqrt{2}}} e^{-t^2} \, dt \tag{24}$$

$$= \frac{\sqrt{\pi}}{\sqrt{2}} erf(\frac{x}{\sqrt{2}})$$

We can continue the computation of the integral further.

$$
\int_a^b \frac{1-x}{\sqrt{2\pi}} e^{-\frac{x^2}{2}} \, dx = \int_a^b \frac{1}{\sqrt{2\pi}} e^{-\frac{x^2}{2}} \, dx - \int_a^b \frac{x}{\sqrt{2\pi}} e^{-\frac{x^2}{2}} \, dx
$$

$$
= \frac{1}{\sqrt{2\pi}} \left( \int_0^b e^{-\frac{x^2}{2}} \, dx - \int_0^a e^{-\frac{x^2}{2}} \, dx - e^{-\frac{a^2}{2}} + e^{-\frac{b^2}{2}} \right) \tag{25}
$$

$$
= \frac{1}{\sqrt{2\pi}} \left[ \left( \frac{\sqrt{\pi}}{\sqrt{2}} erf(\frac{b}{\sqrt{2}}) - \frac{\sqrt{\pi}}{\sqrt{2}} erf(\frac{a}{\sqrt{2}}) - e^{-\frac{a^2}{2}} + e^{-\frac{b^2}{2}} \right] \right.
$$

Equation equation 22 can be written as follows:

$$
\mathbb{E}[|\epsilon_{ij}|] = \frac{2}{\sqrt{2\pi}} \left\{ \left[ \frac{\sqrt{\pi}}{\sqrt{2}} erf(\frac{1}{\sqrt{2}}) - \frac{\sqrt{\pi}}{\sqrt{2}} erf(\frac{0}{\sqrt{2}}) - e^{-\frac{0}{2}} + e^{-\frac{1}{2}} \right] \right.
$$

$$
- \left[ \left( \frac{\sqrt{\pi}}{\sqrt{2}} erf(\frac{\infty}{\sqrt{2}}) - \frac{\sqrt{\pi}}{\sqrt{2}} erf(\frac{1}{\sqrt{2}}) - e^{-\frac{1}{2}} + e^{-\frac{\infty}{2}} \right] \right\} \tag{26}
$$

$$
\xlongequal{erf(0)=0, erf(\infty)=1} 2[erf(\frac{1}{\sqrt{2}}) - \frac{1}{2} - \frac{1}{\sqrt{2\pi}} + \frac{2}{\sqrt{2\pi e}}] \approx 0.5354
$$

Therefore, based on Equations equation 21, the expected value of the additional gradient error $\mathbb{E}[\Delta \frac{\partial L}{\partial w_{mn}^{(l)}}]$ can be expressed as follows:

$$
\mathbb{E}[\Delta \frac{\partial L}{\partial w_{mn}^{(l)}}] \leq \sum_i \sum_j \sum_{m'} \sum_{n'} \mathbb{E}[|\epsilon_{(i+m)(j+n)}^{(l)} \cdot \frac{\partial \sigma(y_{(i+m')(j+n')}^{(l)})}{\partial y_{ij}^{(l)}} \cdot w_{m'n'}^{(l+1)} \cdot \frac{\partial L}{\partial y_{ij}^{(l+1)}}|] \tag{27}
$$

Based on Equation equation 15, since the binarization error $\epsilon_{ij}^{(l)}$ depends solely on the input $x_{ij}^{(l)}$ and is independent of any other variables, $\epsilon_{ij}^{(l)}$ and other random variables in Equation equation 27 are mutually independent. Therefore, it follows that:

$$
\mathbb{E}[\Delta \frac{\partial L}{\partial w_{mn}^{(l)}}] \leq \sum_i \sum_j \sum_{m'} \sum_{n'} \mathbb{E}[|\epsilon_{(i+m)(j+n)}^{(l)}|] \cdot \mathbb{E}[|\frac{\partial \sigma(y_{(i+m')(j+n')}^{(l)})}{\partial y_{ij}^{(l)}} \cdot w_{m'n'}^{(l+1)} \cdot \frac{\partial L}{\partial y_{ij}^{(l+1)}}|]
$$

$$
\approx 0.5354 \cdot \left( \sum_i \sum_j \sum_{m'=-(k//2)}^{k//2} \sum_{n'=-(k//2)}^{k//2} \mathbb{E}[|\frac{\partial \sigma(y_{(i+m')(j+n')}^{(l)})}{\partial y_{ij}^{(l)}} \cdot w_{m'n'}^{(l+1)} \cdot \frac{\partial L}{\partial y_{ij}^{(l+1)}}|] \right) \tag{28}
$$

From the above equations, it is evident that as the size $k$ of the convolutional kernel in the subsequent layer increases, the element-wise gradient error introduced during the binarization process also increases. Consequently, in binarized convolutional units, the smaller the size of the convolutional kernel $k$, the smaller the binarization error introduced into the binarized model.

Therefore, we use $1 \times 1$ binarized convolution as the new binarized convolution.

## A.5 PROOF OF THEOREM 2

**Theorem 2.** *Compared to full-precision convolutional layers, binarized convolutional layers exhibit disadvantages in capturing the scale variations across multiple channels of the feature maps. The specific expression is as follows.*

$$
\sup_{X, \phi_{c_1}, \phi_{c_2}} |S_{\hat{y}^{c_1}} - S_{\hat{y}^{c_2}}| < \sup_{X, \phi_{c_1}, \phi_{c_2}} |S_{y^{c_1}} - S_{y^{c_2}}| \tag{29}
$$

*Let $X \in \mathbb{R}^{C \times H \times W}$ represent the input feature maps, and let $\phi_c$ denote the full-precision convolution kernel of the c-th channel, which satisfies $avg(|\phi_c|) < max(|\phi_c|)$. The term $S$. refers to the scale of the feature map, defined as the normalized $\ell_1$-norm. Furthermore, $y$ and $\hat{y}$ represent the output feature map for a specific channel obtained from $\phi_c$ and its binarized version, respectively.*

**Proof.** We define the input feature maps as $\mathbf{X} = [x^1, x^2, \ldots, x^C], \mathbf{X} \in \mathcal{R}^{C \times H \times W}$, and the output feature maps of the full-precision convolution as $\mathbf{Y} = [y^1, y^2, \ldots, y^C], \mathbf{Y} \in \mathcal{R}^{C \times H \times W}$, where we

assume the number of channels remains unchanged. For the scale $S_{y^c}$ of the $c$-th channel in the output feature map, we have:

$$y^{c,i,j} = \sum_{q=1}^{C} \sum_{m'=-k//2}^{k//2} \sum_{n'=-k//2}^{k//2} (x^{q,i+m',j+n'} \phi_c^{q,m',n'} + b_c^{q,m',n'}) \tag{30}$$

$$S_{y^c} = avg_{i,j}(|y^{c,i,j}|) = \frac{1}{HW} \sum_i \sum_j |y^{c,i,j}|$$

where $\phi_c$ and $b_c$ are the weight and bias of the $c$-th kernel, respectively. Consider $S_{y^{c_1}}$ and $S_{y^{c_2}}$, and if $S_{y^{c_2}} < S_{y^{c_1}}$, we have:

$$S_{y^{c_1}} - S_{y^{c_2}} = \frac{1}{HW} \sum_i \sum_j |y^{c_1,i,j}| - \frac{1}{HW} \sum_i \sum_j |y^{c_2,i,j}| \le \frac{1}{HW} \sum_i \sum_j |y^{c_1,i,j} - y^{c_2,i,j}| \tag{31}$$

Let the bias $b$ be 0, for full-precision convolution:

$$|y^{c_1,i,j} - y^{c_2,i,j}| = |\sum_{q=1}^{C} \sum_{m'=-k//2}^{k//2} \sum_{n'=-k//2}^{k//2} (x^{q,i+m',j+n'} (\phi_{c_1}^{q,m',n'} - \phi_{c_2}^{q,m',n'})|$$
$$\le Ck^2 \cdot max(|x^{q,i+m',j+n'}|) \cdot max(|\phi_{c_1}^{q,m',n'} - \phi_{c_2}^{q,m',n'}|) \tag{32}$$
$$\le Ck^2 \cdot max(|x^{q,i+m',j+n'}|) \cdot (max(|\phi_{c_1}^{q,m',n'}|) + max(|\phi_{c_2}^{q,m',n'}|))$$

For binary convolution, we have:

$$|\hat{y}^{c_1,i,j} - \hat{y}^{c,i,j}| = |\sum_{q=1}^{C} \sum_{m'=-k//2}^{k//2} \sum_{n'=-k//2}^{k//2} (x^{q,i+m',j+n'} (avg(\phi_{c_1}) w_{c_1}^{q,m',n'} - avg(\phi_{c_2}) w_{c_2}^{q,m',n'})|$$
$$\le Ck^2 \cdot max(|x^{q,i+m',j+n'}|) \cdot max(|avg(\phi_{c_1}) w_{c_1}^{q,m',n'} - avg(\phi_{c_2}) w_{c_2}^{q,m',n'}|)$$
$$\le Ck^2 \cdot max(|x^{q,i+m',j+n'}|) \cdot (|avg(\phi_{c_1})| + |avg(\phi_{c_2})|) \tag{33}$$

Here, $w_i^{q,m',n'} = sign(\phi_i^{q,m',n'})$, thus it can be proven that the supremum of $|y^{c_1,i,j} - y^{c_2,i,j}|$ is greater than $|\hat{y}^{c_1,i,j} - \hat{y}^{c_2,i,j}|$. According to equation 31, the supremum of $S_{Y^{c_1}} - S_{Y^{c_2}}$ is greater than or equal to $S_{\hat{y}^{c_1}} - S_{\hat{y}^{c_2}}$. It indicates that binary convolution reduces the scale differences between different feature channels, which implies a decline in attention across feature channels.

### A.6 MORE DETAILS ABOUT EXPERIMENTS

#### A.6.1 RESULT OF DIFFERENT BACKBONE OF BDC

We applied BDC to RenderOcc, with the results shown in Table 6. The performance of our binary model, BDC-RenderOcc, is nearly equivalent to that of the full-precision RenderOcc.

Table 6: **Comparison of the occupancy prediction performance of RenderOcc and BDC-RenderOcc.**

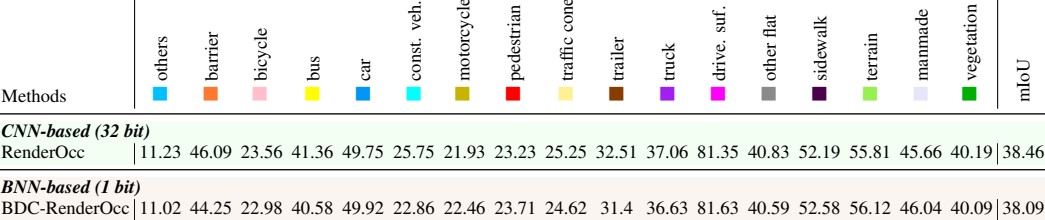

| Methods | others | barrier | bicycle | bus | car | const. veh. | motorcycle | pedestrian | traffic cone | trailer | truck | drive. suf. | other flat | sidewalk | terrain | manmade | vegetation | mIoU |
|---|---|---|---|---|---|---|---|---|---|---|---|---|---|---|---|---|---|---|
| *CNN-based (32 bit)* | | | | | | | | | | | | | | | | | | |
| RenderOcc | 11.23 | 46.09 | 23.56 | 41.36 | 49.75 | 25.75 | 21.93 | 23.23 | 25.25 | 32.51 | 37.06 | 81.35 | 40.83 | 52.19 | 55.81 | 45.66 | 40.19 | 38.46 |
| *BNN-based (1 bit)* | | | | | | | | | | | | | | | | | | |
| BDC-RenderOcc | 11.02 | 44.25 | 22.98 | 40.58 | 49.92 | 22.86 | 22.46 | 23.71 | 24.62 | 31.4 | 36.63 | 81.63 | 40.59 | 52.58 | 56.12 | 46.04 | 40.09 | 38.09 |

#### A.6.2 RESULT OF DIFFERENT VERSION OF BDC

We tested the performance metrics of different versions of BDC on the Occ3d-nuScenes validation set. Table 7 presents the results. The configurations of BDC-B and BDC-T follow the settings

Table 7: **Comparison of the occupancy prediction performance across different versions of BDC.** BDC-S binarizes all modules in the 3D occupancy network except for the view transformer. These modules include an image encoder, BEV encoder, and occupancy head. † stands for not using pre-trained weights from an image backbone.

| Methods | Params(M) | OPs(G) | others | barrier | bicycle | bus | car | const. veh. | motorcycle | pedestrian | traffic cone | trailer | truck | drive. suf. | other flat | sidewalk | terrain | manmade | vegetation | mIoU |
|---|---|---|---|---|---|---|---|---|---|---|---|---|---|---|---|---|---|---|---|---|
| *CNN-based (32 bit)* | | | | | | | | | | | | | | | | | | | | |
| FlashOcc | 44.74 | 248.57 | 9.08 | 46.32 | 17.71 | 42.70 | 50.64 | 23.72 | 20.13 | 22.34 | 24.09 | 30.26 | 37.39 | 81.68 | 40.13 | 52.34 | 56.46 | 47.69 | 40.60 | 37.84 |
| FlashOcc† | 44.74 | 248.57 | 6.10 | 35.78 | 0.50 | 26.97 | 42.39 | 11.16 | 7.13 | 10.99 | 10.68 | 20.95 | 24.35 | 80.60 | 40.02 | 50.44 | 55.11 | 44.67 | 38.85 | 29.81 |
| *BNN-based (1 bit)* | | | | | | | | | | | | | | | | | | | | |
| BDC-T | 26.83 | 129.90 | 10.16 | 44.38 | 18.53 | 41.40 | 49.87 | 23.12 | 20.94 | 22.33 | 23.29 | 29.93 | 36.19 | 81.14 | 39.37 | 51.43 | 55.25 | 47.37 | 40.87 | 37.39 |
| BDC-B | 28.22 | 134.50 | 9.57 | 44.80 | 20.45 | 40.21 | 49.96 | 23.72 | 21.48 | 22.58 | 24.47 | 27.40 | 36.48 | 80.22 | 38.34 | 50.12 | 54.74 | 47.19 | 40.04 | 37.16 |
| BDC-S | 3.51 | 45.30 | 3.13 | 24.25 | 6.02 | 22.21 | 36.23 | 7.29 | 5.78 | 14.11 | 14.04 | 4.86 | 22.99 | 68.21 | 14.29 | 33.52 | 36.76 | 33.20 | 30.63 | 22.21 |

outlined in Table 1. We binarized all modules in the 3D occupancy network except for the view transformer, referring to this as the **small** version (**-S**). These modules include the image encoder, the BEV encoder, and the occupancy head.

Compared to BDC-T, BDC-S additionally binarizes the image backbone in the image encoder. The image backbone contains substantial pre-trained knowledge, and binarizing it hinders leveraging this pre-trained knowledge, which leads to a significant performance drop compared to BDC-T. Compared to FlashOcc†, which does not use pre-trained weights in the image backbone, the binarized version shows a significant performance decline.

Therefore, we recommend against binarizing the image backbone.

### A.6.3 PERFORMANCE OF BINARIZED MODULE OF 3D OCCUPANCY NETWORK

We binarized different modules in the occupancy network. The following table reports the mIoU of binarizing different modules.

Table 8: Model Performance Metrics

| | Only Image Neck | Only BEV Backbone | Only BEV Neck | Only Occupancy Head | BDC-T | FlashOcc |
|---|---|---|---|---|---|---|
| mIoU | 37.91 (+0.07 ↑) | 31.62 (-6.22 ↓) | 37.59 (-0.25 ↓) | 31.46 (-6.38 ↓) | 37.39 (-0.45 ↓) | 37.84 |

According to Table 8, we can find: (1) The binarization of the BEV backbone and the occupancy head significantly impacts performance. (2) During joint training, the binarization errors of the entire network can be considered and optimized as a whole.

### A.6.4 OPERATIONS AND PARAMETERS OF BINARIZED MODULE OF 3D OCCUPANCY NETWORK

In Table 9, we investigate the changes in computation (OPs) and parameters (Params) across different modules of the 3D occupancy network before and after binarization. The image encoder consists of the image backbone and image neck, while the BEV encoder includes the BEV backbone and BEV neck. (x%) indicates that x% of the full-precision operations/parameters have been binarized.

We do not binarize the view transformer because its 32-bit full-precision parameters and computation are already sufficient. Additionally, the view transformer relies on full-precision computation to precisely map 2D image features to 3D BEV features.

Table 9: **The proportion of 32-bit OPs and Params versus 1-bit OPs and Params in each module of BDC-T.** (%) denotes the proportion of 32-bit and 1-bit operations within each module.

|  | Model | Bit | Image Backbone | Image Neck | View Transformer | BEV Backbone | BEV Neck | Occupancy Head | Total |
|---|---|---|---|---|---|---|---|---|---|
| OPs(G) | FlashOcc | 32-bit | 88.785 | 1.377 | 0.165 | 17.724 | 102.989 | 34.755 | 248.572 |
|  | BDC-T | 32-bit | 88.785 | 0 | 0.165 | 0 | 29.491 (28.64%) | 11.141 (32.06%) | 129.582 (52.13%) |
|  |  | 1-bit | 0 | 0.034 | 0 | 0.046 | 0.474 (71.36%) | 0.031 (67.94%) | 0.585 (47.87%) |
| Params(M) | FlashOcc | 32-bit | 23.508 | 4.155 | 0.039 | 12.394 | 6.556 | 0.869 | 44.744 |
|  | BDC-T | 32-bit | 23.508 | 0 | 0.039 | 0 | 2.949 (44.98%) | 0.279 (32.11%) | 26.775 (59.84%) |
|  |  | 1-bit | 0 | 0.022 | 0 | 0.020 | 0.012 (55.02%) | 0.001 (67.89%) | 0.055 (40.16%) |

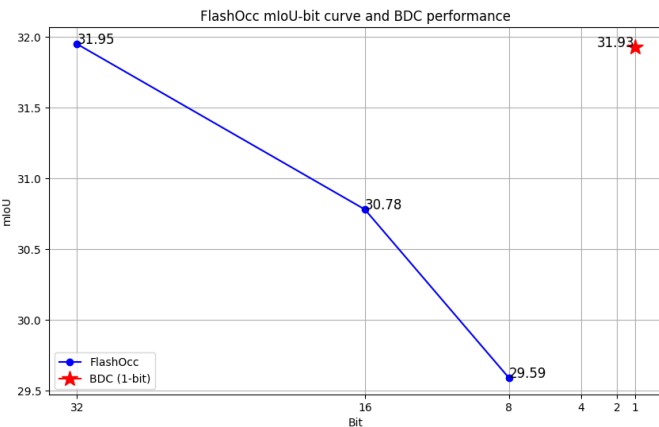

Figure 9: FlashOcc mIoU-bit curve and BDC performance

### A.6.5 MIOU-BIT CURVE VISUALIZATION

We used FlashOcc without temporal information as the baseline and applied the BDC-T method for binarization on this baseline. We then plotted the performance of FlashOcc models at different bit levels and compared it with the performance of BDC-T.

As shown in Figure 9, the performance of our BDC-T is comparable to that of the full-precision model and superior to the performance of FlashOcc at both 16-bit and 8-bit levels.

### A.6.6 MORE VISUALIZATION

In this section, we provide additional occupancy prediction results of BiSRNet Cai et al. (2024) and our BDC applied to Flashocc in Fig 10. Compared to BiSRNet, BDC offers superior scene reconstruction capability and more accurate label prediction.

### A.7 BROADER IMPACTS

3D occupancy prediction stands as a core task in autonomous driving perception. Leveraging occupancy grids effectively address real-world challenges such as long-tail datasets and target truncation, which 3D object detection algorithms may struggle to resolve. Our approach, BDC-Occ, demonstrates superior efficiency and accuracy in predicting the occupancy status of voxels in 3D space compared to all existing state-of-the-art methods based on Binarized Neural Networks (BNNs),

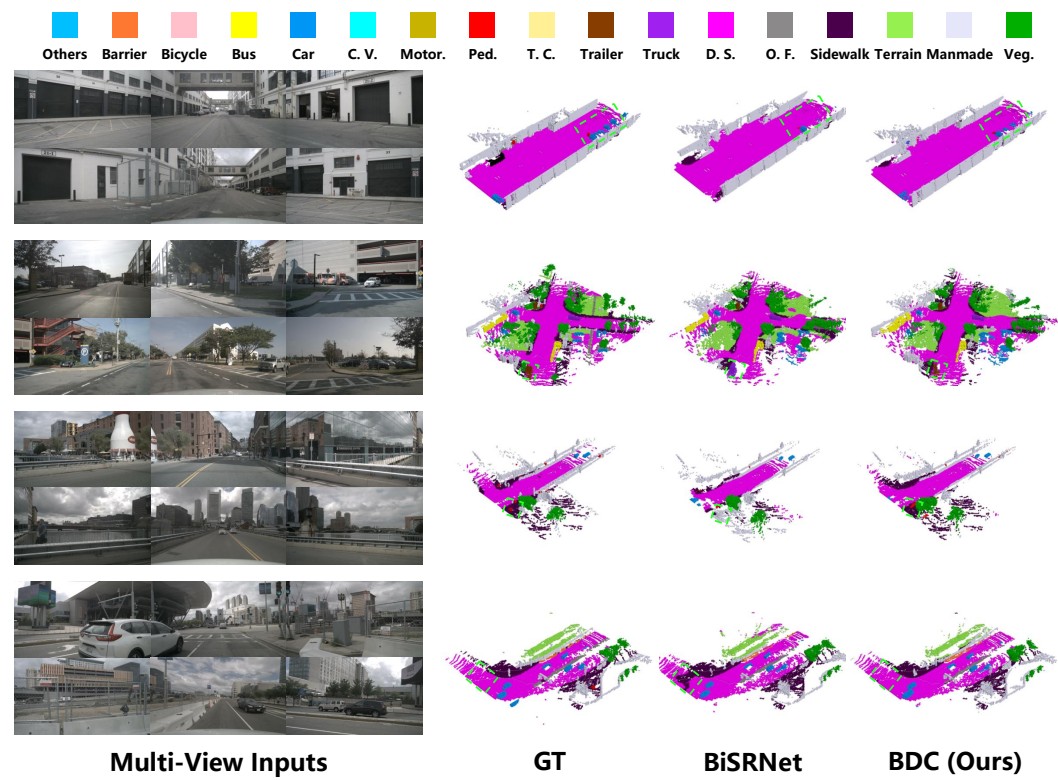

Figure 10: More Visualization rensults on Occ3D-nuScenes validation set

holding significant value for practical applications. Thus far, 3D occupancy prediction technology has not yielded any adverse societal impacts. Our proposed BDC-Occ likewise does not introduce any foreseeable negative social consequences.

