# OpenReview forum: "BDC-Occ: Binarized Deep Convolution Unit For Binarized Occupancy Network"
_ICLR.cc/2025/Conference — ICLR 2025 Conference Withdrawn Submission_

### Official Review · Reviewer_c8JS · 2024-10-27

**Soundness:** 3
**Presentation:** 3
**Contribution:** 2
**Rating:** 5
**Confidence:** 2

**Summary:**

To reduce parameters and computational costs, this paper proposes a binarized occupancy network tailored specifically for CNN-based occupancy networks. The approach introduces two key techniques: first, an additional 1x1 binarized convolution layer is added to increase network depth, thereby enhancing feature extraction while maintaining efficiency. Second, a per-channel refinement branch is incorporated to reduce quantization error, improving the model’s precision despite the constraints of binarization. These enhancements aim to optimize the balance between performance and resource efficiency in the proposed binarized network.

**Strengths:**

Occupancy estimation plays a critical role in real-time applications like autonomous driving, where precise environment mapping is essential for safe and efficient vehicle navigation. However, the computational demands of occupancy estimation, especially when deployed on device-side platforms like Orin GPUs, create significant challenges in terms of efficiency and resource constraints. This research addresses these challenges by exploring strategies to minimize computational overhead, a vital area of study given the increasing emphasis on edge computing in autonomous systems. By carefully engineering these binarization blocks, the proposed approach effectively mitigates quantization loss, thus maintaining model accuracy while reducing resource consumption.

**Weaknesses:**

# Major Concern:
While this work serves as an early exploration of binary quantization for the occupancy prediction task, several significant limitations are apparent. The study does present two main findings in the context of binary occupancy networks:
- 1×1 binarized convolution introduces only minimal binarization errors as network depth increases.
- Binarized convolution is notably less effective than full-precision convolution at capturing cross-channel feature importance.

However, the proposed strategies to address these issues—namely, enhancing network depth through additional 1x1 binarized layers and adding a per-channel refinement branch—have already been extensively explored in the quantization literature. This repetition limits the novelty of the work, and the paper does not provide fresh insights into the specific task of occupancy prediction. The binary quantization methods introduced here, while perhaps incrementally improving upon existing binary quantization techniques, do not directly tackle the occupancy task itself. Instead, they represent a basic application of binary quantization to occupancy, yielding largely expected results. As such, I believe this work falls short of the standards expected at a top-tier machine learning conference, and the main track may not be the optimal venue for this submission.

# Minor Comments:
- This study restricts its exploration to CNN-based approaches for occupancy tasks, without incorporating transformer-based occupancy networks. Transformer models have gained traction and demonstrated superior performance in related fields, and their absence here limits the study’s relevance to current research trends.
- The paper outlines a manually crafted binary network architecture aimed at mitigating quantization-induced errors. However, quantizing both weights and activations from 32-bit to 1-bit introduces significant quantization errors. Without innovations specifically targeted at reducing such errors in this context, the proposed method struggles to demonstrate substantial improvements.
- The speed advantage observed (6.22 FPS versus 7.53 FPS) is marginal, casting doubt on the practical benefits and motivation of the approach. If the goal is to address computational cost and memory constraints in existing occupancy networks, a rigorous analysis of actual speedup and memory reduction should be provided. Alternatively, if the study’s focus is to explore novel quantization techniques for this task, one would expect groundbreaking insights tailored to occupancy prediction. Given the current submission, both aspects seem underdeveloped.

**Questions:**

- The paper would benefit from a clearer description of the hardware platform used and the specific implementation strategy for 1-bit quantized convolution. Key questions remain unanswered, such as whether the quantized operations are run on a specialized computing framework like TensorRT, which is widely used for deploying optimized deep learning inference on edge and server devices. Additionally, details on how 1-bit quantized convolution is implemented—such as the use of custom kernels, acceleration libraries, or optimizations for reduced precision—would provide readers with a clearer understanding of the practical feasibility and performance considerations of this approach.

- If the work relies on standard hardware (e.g., CPUs or GPUs) without the assistance of dedicated quantization frameworks, this would impact performance significantly compared to using more specialized hardware like TPUs or FPGAs, which are often better suited for extreme quantization levels. Specifying these factors is crucial for evaluating the performance claims, as well as the actual benefit of 1-bit quantization in terms of speed and efficiency.

---

> ### Comment · Reviewer_c8JS · 2024-11-27
>
> Given the lack of response, I am assuming the authors have decided not to proceed with addressing the required revisions or feedback. As such, I would like to conclude the review process for this submission.

---

### Official Review · Reviewer_yPtU · 2024-10-29

**Soundness:** 3
**Presentation:** 3
**Contribution:** 3
**Rating:** 3
**Confidence:** 3

**Summary:**

BDC-Occ proposes a BDC unit that applies binarization to 3D occupancy prediction tasks. To alleviate binarization errors, they use a 1x1 binarized convolution and design a BDC unit based on it. The authors show that their method reduces the performance gap with floating-point models and significantly reduces hardware resources.

**Strengths:**

1. This paper is the first study to apply binarization to the task of 3D occupancy prediction. The method of the authors significantly reduces the computational cost while maintaining the performance.
2. The proposed BDC unit significantly improves the performance of the network through theoretical analysis.

**Weaknesses:**

1. what is the contribution of BDC-V2 in Figure 3? It seems that it only increases the computational cost, with no performance improvement. Furthermore, MultiBiconv seems to be a multiple iteration of the technique from BDC-V1.
2. The paper seems to propose the design of a binarization methodology, not a design for Occupancy Prediction. While this is the first application of binarization to occupancy prediction, other binarization methodologies seem to be easily adaptable.
3. The design in Section 3.4 seems to be very similar to that of BiSRNet, which may limit the contribution of the paper. Is there anything different about the design of BiSRNet?

**Questions:**

1. In the paper, binarization is adopted for deployment on edge devices. Is there any experiment related to this?
2. The FPS in Table 3 seems to be a small improvement compared to FlashOcc, which is a floating point model. Is there any experimental results of FPS and run time of BDC-B?

---

### Official Review · Reviewer_moDb · 2024-11-03

**Soundness:** 3
**Presentation:** 2
**Contribution:** 3
**Rating:** 3
**Confidence:** 3

**Summary:**

This paper introduces BDC-Occ, a novel binarized neural network (BNN) for 3D occupancy prediction. Recognizing the computational challenges of deploying existing 3D occupancy networks on edge devices, the authors leverage BNNs for model compression. However, they note that simply binarizing existing models leads to performance degradation, particularly when increasing network depth.

The paper's core contribution is the Binarized Deep Convolution (BDC) unit.  It addresses the identified limitations of binarized convolutions through two key innovations. First, additional convolutional kernels within the BDC are constrained to 1x1 to minimize the impact of binarization errors as network depth increases. Second, a per-channel refinement branch reweights outputs using a first-order approximation, improving the capture of cross-channel feature importance.

Through extensive experiments on the Occ3D-nuScenes dataset, the authors demonstrate that BDC-Occ achieves state-of-the-art performance among BNNs, even rivaling full-precision models in mIoU while significantly reducing parameters and operations.  They further validate the generalizability of the BDC unit by showing its effectiveness in 3D object detection tasks on the nuScenes dataset.

**Strengths:**

Originality: The paper demonstrates originality in its identification and solution for the performance degradation problem in binarized 3D occupancy networks. While BNNs have been explored in other domains, applying them effectively to 3D occupancy prediction, especially with a focus on maintaining performance with increasing network depth, is novel. The proposed BDC unit, with its 1x1 kernel constraint and the per-channel refinement branch, presents a creative combination of techniques tailored to address the specific challenges of binarization in this context.

Quality: The technical quality of the paper is good. The authors provide theoretical justification for their design choices and conduct thorough experiments to validate the effectiveness of the BDC unit. The results convincingly demonstrate the superiority of BDC-Occ over other state-of-the-art binarized methods and its competitiveness with full-precision models. The ablation studies further strengthen the claims by showcasing the individual contributions of each component of the BDC unit.

Clarity: While the core ideas are presented clearly, the clarity of the paper could benefit from some improvements. The mathematical proofs are clear. Additionally, visually presenting the overall architecture of BDC-Occ aid understanding.

Significance: The significance of the work lies in its potential to enable deployment of accurate 3D occupancy prediction on edge devices. The substantial reduction in computational cost achieved by BDC-Occ, without sacrificing accuracy, is notable. It would be nice to discuss the roadmap to apply on transformer, to inspire further researches.

**Weaknesses:**

1. **Limited exploration of binarization strategies**: The paper primarily focuses on binarizing convolutional layers using the BiSR-Conv method.  Exploring alternative binarization techniques, such as XNOR-Net [1] or DoReFa-Net [2], and comparing their performance with BDC-Occ would strengthen the analysis and potentially reveal further insights.  It's also important to investigate the impact of different activation functions specifically designed for BNNs, like those proposed in ReactNet [3].

2. **[minor] Lack of comparison with other compression techniques**: The paper positions BDC-Occ as a solution for deploying occupancy networks on edge devices. However, it lacks a comparison with other model compression methods beyond binarization, such as pruning [4], quantization [5], or knowledge distillation [6].  Demonstrating the advantages of BDC-Occ over these alternatives would significantly bolster its practical significance.

3. **[minor] Limited generalizability**: The paper acknowledges the limitation to CNN architectures. However, this limitation needs further discussion and investigation. This would help understand the challenges and potentially open avenues for future research on binarizing more diverse network architectures.


**References:**

[1] Hubara et al., Binarized Neural Networks, NIPS 2016.

[2] Zhou et al., DoReFa-Net: Training Low Bitwidth Convolutional Neural Networks with Low Bitwidth Gradients, arXiv 2016.

[3] Liu et al., Reactnet: Towards precise binary neural network with generalized activation functions, ECCV 2020.

[4] Han et al., Learning both Weights and Connections for Efficient Neural Networks, NIPS 2015.

[5] Jacob et al., Quantization and Training of Neural Networks for Efficient Integer-Arithmetic-Only Inference, CVPR 2018.

[6] Hinton et al., Distilling the Knowledge in a Neural Network, arXiv 2015.

**Questions:**

Here are some questions and suggestions for the authors:

1. **Alternative Binarization Techniques**: The paper focuses on BiSR-Conv for binarization. Could the authors provide a comparison with other established techniques like XNOR-Net or DoReFa-Net? This would help understand if the performance gains stem specifically from the BDC unit or are also achievable with other carefully tuned binarization methods.  Furthermore, exploring ternary or higher bit-width quantization could provide a more nuanced understanding of the trade-off between accuracy and efficiency.

2. **[minor] Comparison with Other Compression Methods**:  Beyond BNNs, how does BDC-Occ compare to other model compression techniques like pruning, quantization, and knowledge distillation in the context of 3D occupancy prediction?  Providing quantitative results or discussion comparing these methods would strengthen the argument for BDC-Occ's practical value.

3. **Detailed Analysis of Per-Channel Refinement Branch**:  The ablation study provides some insights, but a more in-depth analysis of the per-channel refinement branch is needed.  How sensitive is its performance to the number of layers in `MulBiconv` (N)? Are there alternative architectures for this branch that could further improve cross-channel feature learning? A visualization of the learned channel weights might also offer insightful qualitative analysis.

4. **[minor] Generalizability to Transformers**: The stated limitation to CNN architectures raises questions about BDC's broader applicability. Have the authors explored applying BDC, or a modified version, to Transformer-based occupancy networks?  Even negative results in this direction would be valuable for understanding the challenges and potential future work.

---

> ### Comment · Reviewer_moDb · 2024-12-02
>
> My questions are not answered by a rebuttal, reject explicitly

---

### Official Review · Reviewer_Gj8y · 2024-11-03

**Soundness:** 2
**Presentation:** 3
**Contribution:** 2
**Rating:** 5
**Confidence:** 4

**Summary:**

This paper presents a method for binarizing convolution operations in binary occupancy networks. It first analyzes the impact of binarization theoretically, and then proposes a mitigation approach based on binarized convolution unit that enhances performance. It provides results based on two benchmarks and comparisons with other networks and binarization methods.

**Strengths:**

- It shows as main strength the theoretical insights that 1x1 binarized convolution is more robust to binarization and is thus used in to make the network deeper. Furthermore, it introduces an additional branch within the network to further refine the per-channel output of each layer based on this observation.
- The results indicate an improvement over other binarization methods in terms of IoU and mAP.
- Ablation studies are performed to show the impact of the various proposed changes.

**Weaknesses:**

- In terms comparing the FPS with FlashOCC, one of the main objectives of binarization is to provide FPS speedups, since the provided results are marginally improved at best it seems to make the proposed approach less necessary. Instead FPS should be compared with other binarization methods as well for fairness.
- From the current writeup it is not clear how the proposed module can be plugged into other existing methods.
- I am unconvinced that since in my understanding the backbone is left as FP32 why binarization is necessary and other forms of quantization are not considered appropriate.
- It is not clear what aspects of the approach are specific to the occupancy task and what is generalizable to other tasks as well.

**Questions:**

Why are so many versions of the main block are needed?

---

> ### Comment · Reviewer_Gj8y · 2024-11-26
>
> I have not seen a rebuttal for this paper so my initial rating stands.

---

### Note · Authors · 2024-12-05

I have read and agree with the venue's withdrawal policy on behalf of myself and my co-authors.